# Effect of Roasting Level on the Development of Key Aroma-Active Compounds in Coffee

**DOI:** 10.3390/molecules29194723

**Published:** 2024-10-06

**Authors:** Andrea M. Obando, Jorge G. Figueroa

**Affiliations:** Departamento de Química, Universidad Técnica Particular de Loja, San Cayetano Alto s/n, Loja 1101608, Ecuador

**Keywords:** roasted coffee, aroma-active compounds, SPME, GC-O, GC-MS

## Abstract

Coffee roasting is considered the most critical process in the development of sensory characteristics. During this stage, a substantial number of compounds are generated. Nevertheless, only a limited number of these compounds are responsible for the aroma, referred to as key aroma-active compounds. This study aimed to assess the impact of roasting levels on the formation of these compounds. Coffee was roasted at four different levels: very light (RL85), light (RL75), medium (RL55), and extremely dark (RL25), according to the Specialty Coffee Association (SCA) guidelines. The extraction, olfactory evaluation, and identification of compounds were performed using solid-phase microextraction (SPME), gas chromatography–olfactometry (GC-O), and gas chromatography–mass spectrometry (GC-MS), respectively. A total of 74 compounds were successfully identified, of which 25 were classified as aroma-active compounds. RL75 and RL85 displayed similar aromatic profiles. RL55 was characterized by pleasant notes such as sweet, toasted hazelnut, and caramel. In contrast, RL25 was marked by undesirable odors including burnt notes, putrid, and sulfurous scents. This study is the first to identify key odorant compounds in coffee based on SCA roasting standards.

## 1. Introduction

Coffee is one of the most widely consumed beverages globally, valued for its pleasant aroma [1]. The aroma depends on multiple factors: intrinsic factors, such as the coffee species and variety, among others [2]; and extrinsic factors, including fermentation, drying, roasting, grinding, storage, and brewing methods [2,3]. Nevertheless, the roasting process is considered the critical point, as it involves multiple chemical reactions such as the Maillard reaction, caramelization, and Strecker degradation, among others [4,5]. These reactions produce a large number of chemical compounds. Additionally, the roasting level influences the generation of different classes and concentrations of chemical compounds, which can significantly affect the color, flavor, and aroma of the coffee [6]. In this regard, medium roasting is commonly used for the commercialization of high-quality coffees, typically with an RL55 value according to the SCA scale [7]. Conversely, a more intense roast (<RL25) is often employed to mask defects in the coffee or adulterations [8].

Volatile compound extraction can be accomplished using various analytical techniques, such as headspace, simultaneous distillation extraction, supercritical fluid extraction, steam distillation, vacuum extraction with water or organic solvents, liquid–liquid extraction, and SPME, among others [9,10,11]. This extraction technique has been widely employed in the analysis of various foods, including roasted coffee and coffee beverages [2,12,13,14,15]. SPME is characterized by being an economical, rapid, solvent-free method that requires minimal sample handling [2,10]. Furthermore, this extraction technique is compatible with GC-MS and GC-O.

Regarding the identification and quantification of compounds, GC is the primary technique used [9,16]. The detectors commonly employed in coffee analysis include Flame Ionization Detection (FID) and Mass Spectrometry (MS) [17]. These powerful detectors facilitate the identification and quantification of chemical compounds. However, they do not enable the recognition of compound odors. In this context, the combination of GC-O has enabled analysts to identify the odor of each compound after separation, and importantly, to determine whether the odor is pleasant or unpleasant [2,18,19,20].

In coffee, over 1000 compounds from various chemical classes have been identified [21]. Nonetheless, several studies have found that only a small group of compounds is responsible for the aroma of the coffee [6,12,13]. These compounds are known as key aroma-active compounds [22]. Blank et al. [23] identified 13 compounds as significant contributors to the aroma of roasted coffee powder. Similarly, Mayer, Czerny, and Grosch [22] found that the aroma of coffee brew was primarily due to certain alkylpyrazines, furanones, and phenols, as well as 2-furfurylthiol, methional, and 3-mercapto-3-methylbutyl formate. Yet these studies employed extraction methods that could generate compounds through reactions between the coffee’s inherent compounds and the reagents used in extraction [17,22,23]. Moreover, López-Galilea, Fournier, Cid, and Guichard [2] used SPME-GC-O to identify 34 compounds with high odor impact in espresso or filtered coffee. All these studies have assessed only a single roasting level. Conversely, a recent study evaluated the effect of roasting level on the generation of key aroma-active compounds; nonetheless, the roasting conditions assessed were not compatible with the commercialization of specialty coffees [24]. Given the importance of roasting in the formation of aroma-related compounds and the lack of studies, the present study aimed to identify the effect of coffee roasting on the formation of key aroma-active compounds. The findings will contribute to improving post-harvest processes to maximize the concentration of compounds with pleasant aromas.

## 2. Results and Discussion

### 2.1. Identification of Volatile Compounds

Using SPME-GC-MS, 74 compounds were identified (Appendix A). Nevertheless, only 25 volatiles were found to impact the aroma in the olfactometric analysis. Table 1 presents the linear retention indices calculated for the DB-5MS and DB-WAX columns, along with their descriptions and odor intensities. The identified aroma-active compounds were classified into ten categories: eight furans, seven pyrazines, two ketones, one thiazole, one ester, one hydrocarbon, one alcohol, one pyran, and one pyrrole, with one compound remaining unidentified. It is noteworthy that the volatiles exhibiting aroma have been previously identified in various studies [9,13,23].

Quantitatively, the two main chemical classes present in coffee are furans and pyrazines (Figure 1). This finding is consistent with the observations of Angeloni, Mustafa, Abouelenein, Alessandroni, Acquaticci, Nzekoue, Petrelli, Sagratini, Vittori, and Torregiani [4], who noted that these compound groups significantly contribute to the flavor and aroma of coffee. As illustrated in Figure 1, the intensity of furan compounds exhibits significant variability. 2,5-dimethylfuran (compound **4**) exhibited a roasted poultry aroma, with consistent intensity across all roast levels. Additionally, 2-furanmethanol (compound **7**), characterized by unpleasant roasted notes, was detected at all roasting levels, with the highest intensity observed at RL75.

The compounds furfuryl formate (compound **9**), 1-(2-furanyl) ethanone (compound 10), and 2,4-dihydroxy-2,5-dimethyl-3(2*H*)-furan-3-one (compound **13**) were also noted for their unpleasant aromas. These compounds exhibited an odor resembling blackcurrant. Regarding its concentration, furfuryl formate was only detected in RL25, whereas the other two compounds exhibited their highest concentrations in RL55. Moreover, the compound 2-acetyl-5-methylfuran (compound **19**) exhibited burnt toasted characteristics; however, it was only detected olfactometrically at the over-roasted level (RL25). Additionally, an increase in concentration was observed at the RL55 and RL25 levels. The compound 4-hydroxy-2,5-dimethyl-3(2*H*)-furanone (compound **20**) was perceived as having a very pleasant caramel odor, with consistent intensity across all evaluated roast levels. The compound 2-furfuryl-5-methylfuran (compound **25**) was noted for its intense green leaf odor, with higher intensity detected at RL25.

Other significant contributors to the aroma of roasted coffee were the pyrazine family. Among the identified pyrazines, 2,5-dimethylpyrazine (compound **11**) exhibited a roasted coffee aroma. Olfactometric analysis revealed that this odor was sensorily more intense in the RL75 roast level (Table 1). For ethenylpyrazine (compound **12**), toasted bread notes were detected, with its concentration decreasing in the RL25. The compounds 2-ethyl-6-methylpyrazine (compound **15**) and 2-ethyl-3-methylpyrazine (compound **16**) were described as having roasted hazelnut and roasted nutty aromas, respectively. However, these compounds were not olfactometrically detected in the RL25. 2-(n-propyl) pyrazine (compound **17**) exhibited an unpleasant rubber-like odor, detected only in the RL25 and RL55 roast levels. According to Chin, et al. [42], this analyte is associated with sulfur, smoky, and bean-like odors. Furthermore, an increase in the concentration of this compound was observed in relation to the roast level. 2-ethenyl-6-methylpyrazine (compound **18**) was noted for its hazelnut aroma, with higher intensity observed in RL55. Finally, 3-ethyl-2,5-dimethylpyrazine (compound **21**) was detected in olfactometric analysis only at the RL75 and RL85 roast levels, with a characteristic roasted hazelnut aroma.

Ketones impart sweet, creamy, and buttery attributes [42]. Among the ketones identified were 2,3-butanedione and 2,3-pentanedione (compounds **2** and **3**), which are responsible for generating sweet aromas. Compound **2** varied from a mild sweet to a burnt aroma at RL25 and RL55, respectively. In contrast, compound **3** was olfactometrically detected only at RL55 (Table 1).

Finally, several compound families were represented by only one member each. The compound 4-methylthiazole (compound **5**) was characterized by an unpleasant, foul odor, detected olfactometrically and chromatographically solely at the RL25 level. Similarly, the compound 2-methylpyridine (compound **6**) was associated with a sulfurous odor. The ester 1-(acetyloxy)-2-propanone (compound **8**) was perceived to have medicinal notes, while β-myrcene (compound **14**) emitted an aroma reminiscent of blackcurrant. Both linalool (compound **22**) and maltol (compound **23**) exhibited sweet notes. Compounds **5**, **6**, **8**, **14**, **22**, and **23** showed an increase in their concentration corresponding to the roast level. Furthermore, 1-furfuryl-2-methylpyrrole (compound **24**) presented a crushed plant odor, with equal intensity at the RL85 and RL55 levels. Lastly, the unidentified compound (compound **1**) was perceived as having a rotten potato aroma and was detected olfactometrically only at the RL25 level.

Compounds **1**, **5**, and **9** were detected olfactometrically exclusively at the RL25 level. Overall, at this roasting level the perceived aromas were unpleasant, characterized by notes of burnt sugar, unpleasant (fetid) odors, sulfurous scents, and burnt coffee, among others.

In the study by Yang, Liu, Liu, Degn, Munchow, and Fisk [13] on aromatic compounds present in coffee, they identified and described similar odors for the compounds 2,5-dimethylfuran, 2,5-dimethylpyrazine, 2-ethyl-6-methylpyrazine, 2-ethenyl-6-methyl pyrazine, and maltol. Similarly, Chin, Eyres, and Marriott [42], in their research on impactful compounds in roasted coffee, agreed with the aromas of 2,3-butanedione, 2,3-pentanedione, 2-furanmethanol, 2-(n-propyl) pyrazine, and 2-acetyl-5-methylfuran. Likewise, the aroma profiles of compounds such as 2,5-dimethylfuran, 4-hydroxy-2,5-dimethyl-3(2*H*)-furanone, 1-furfuryl-2-methylpyrrole, 2-ethyl-3-methylpyrazine, 3-ethyl-2,5-dimethylpyrazine, linalool, 2-methylpyridine, 2-furfuryl-5-methylfuran, and ethenyl pyrazine align with the descriptions of coffee aroma reported in previous studies [5,23,43,44].

Conversely, there were four aromatic compounds whose aroma profiles did not match those described in prior studies. Flament [44] describes 1-(acetyloxy)-2-propanone as having a buttery, somewhat sour aroma, β-myrcene as having a balsamic, resinous scent, and 4-methylthiazole as having tomato, fruity, and green notes. Nevertheless, specific details regarding the extraction methods for these volatiles were not available, hindering the comparison of extraction methods used in this study with those applied in sensory analyses. Additionally, furfuryl formate has been identified with a pleasant odor from a mixture of synthetic products [45]. Even so, synthetic preparation might influence sensory perception [46], in contrast to using the roasted coffee sample directly. Finally, the compound 2,4-dihydroxy-2,5-dimethyl-3(2*H*)-furan-3-one has not been described in previous research.

Figure 2a,b present the chromatograms for RL55 and RL25 levels, respectively. It can be observed that although the peak areas for most olfactometrically detected compounds were low, these compounds had an impact on the aroma. Conversely, there were peaks with larger areas, such as acetic acid (A), 1-(acetyloxy)-2-propanone, 1-hydroxy (B), pyridine (C), 2-methyl-3(2*H*)-furanone (D), 3-furaldehyde (E), 5-methyl-2-furancarboxaldehyde (F), and 2-furanmethanol acetate (G), which did not present any aroma in the sensory analysis of this study. These compounds have been previously identified by Flament [44], who described various compounds and their odors. Compound A has sensory characteristics of acrid, sour, and unpleasant. Compound B has a slightly caramelized acrid and sweet odor. Compound C is associated with unpleasant odors. Compound F has been found to have a decreasing aroma with increasing roasting temperature and time. Although RL25 was a dark roast, the aromas of these compounds could not be identified because the maximum temperature used was 195 °C, whereas these volatiles typically require temperatures exceeding 230 °C during roasting. Compound E is described as having a sweet roasted aroma, with a sweet-spicy and slightly caramelized scent. Compound G may produce a soft, ethereal-floral odor. Finally, no specific aroma for compound D has been established in coffee.

### 2.2. Principal Component Analysis (PCA)

Given the diversity of aromatic compounds present in coffee, visualizing their distribution across different roasting levels is challenging. Therefore, PCA was employed to identify the relationship between roasting levels and the presence of aroma-active compounds. For this analysis, data obtained from the FID detector used in the GC-O analysis were utilized. Figure 3 consolidates all the obtained data into two principal axes. The first and second principal components (PC1 and PC2) account for 58% and 27% of the total variance, respectively.

A clear separation is evident between key aroma-active compounds and roasting levels. In the red rectangle, compounds that are prominent at the RL55 level are grouped, including ethenyl pyrazine, 2-ethyl-6-methylpyrazine, 4-hydroxy-2,5-dimethyl-3(2*H*)-furanone, and 2,3-pentanedione, among others. These compounds are characterized by sweet notes, roasted hazelnut, toasted bread, and caramel (pleasant odors), and should be highlighted to enhance the sensory quality of the coffee. On the other hand, the green rectangle displays compounds associated with RL25, featuring furfuryl formate, 4-methylthiazole, 2,3-butanedione, maltol, 2-furfuryl-5-methylfuran, linalool, 2-(n-propyl) pyrazine, β-myrcene, 2-methylpyridine, 2-acethyl-5-methylfuran, 1-(acetyloxy)-2-propanone, and an unidentified compound. These compounds have unpleasant, bitter, and burnt aromatic notes. Finally, the blue rectangle highlights compounds associated with RL85 and RL75, such as 2,5-dimethylpyrazine, 2,5-dimethylfuran, 2-furanmethanol, 3-ethyl-2,5-dimethylpyrazine, and 2-ethyl-3-methylpyrazine, which exhibit a similar aromatic profile.

These results align with the findings of Gonzalez-Rios et al. [47], who indicated that a light roast is characterized by pleasant, fruity, and nutty notes, whereas a dark roast presents unpleasant, burnt, and pungent sensory attributes. It should be noted that RL25 represents a very dark roast, where the Maillard reaction, when subjected to extremely high roasting temperatures, decomposes acids, burns, and carbonizes sugars, generating compounds that are undesirable in coffee aroma. Moreover, these compounds can be classified as highly hazardous to health due to their carcinogenic nature. In contrast, very light roasts (RL75 and RL85) do not allow for the full development of aromatic profiles. Therefore, it is recommended to control the roasting process to achieve an RL55 level, which will facilitate the generation and retention of most desirable coffee compounds [6,47,48].

### 2.3. Heat Map

To analyze the key aroma-active compounds present across the four coffee roasting levels, a heat map was generated (Figure 4) using the peak areas from the GC-O analysis. This representation utilizes a color scale to facilitate the rapid visualization of each aromatic compound present at different roasting levels. The evaluated samples were clearly divided into two groups based on the roasting level: one group included the lightly roasted samples (RL75 and RL85), while the other group comprised the more heavily roasted samples (RL25 and RL55). The distribution of volatile compound concentrations is evident. In RL55, compounds such as ethenyl pyrazine, 4-hydroxy-2,5-dimethyl-3(2*H*)-furanone, and 2-ethyl-6-methylpyrazine are observed, representing roasted, caramel, and toasted hazelnut aromas, respectively.

Conversely, RL75 and RL85 exhibit similar coloration among the compounds. For example, 2-furanmethanol (unpleasant), 2,5-dimethylpyrazine (roasted coffee), and 3-ethyl-2,5-dimethylpyrazine (toasted nut) show comparable colors. Finally, RL25 is characterized by high coloration (intense red) for compounds such as the unidentified compound, 4-methylthiazole, and furfuryl formate, which have unpleasant odors and are present only at this roasting level.

## 3. Materials and Methods

### 3.1. Sample

Samples of coffee (*Coffea arabica* L.) of the Catimor variety (a generic designation derived from the Caturra and Timor Hybrid varieties) [49] were analyzed. These samples were collected from the farm owned by Bolívar Troya, located at 1200 m above sea level in the Palanda canton, Zamora-Chinchipe province, Ecuador. The cherries were manually washed with abundant tap water to remove impurities, dirt, and defective seeds. Afterward, the cherries were immersed in water, and floating fruits were discarded. The coffee cherries were then pulped using a pulping machine (DH-2, Penagos, Bucaramanga, Colombia). Subsequent, 900 g of green coffee seeds was placed in 1000 mL glass bottles, which were sealed with an air trap. The samples were fermented for 72 h at room temperature.

### 3.2. Drying, Hulling, Roasting, and Milling of Coffee

The green coffee seeds were dried using air at a temperature of 50 °C. A tray-type dehydrator with hot air circulation (DY-110H, Lassele, Ansan, Republic of Korea) was used until the seeds reached a moisture content of 10–12% [9]. Once the optimal moisture level was achieved, the parchment was removed using a hulling machine (ING-C-250, Bogotá, Colombia). Four roasting levels (RL) were evaluated, corresponding to very light (RL85), light (RL75), medium (RL55), and extremely dark (RL25) disks, according to the SCA scale [7], as shown in Figure 5. The coffee seeds were roasted according to the SCA protocol [50]. Briefly, the roaster (TC 300 A R/G, Quimbaya, Quantik, Colombia) was preheated to a temperature of 160 °C and maintained at this temperature for 3 min. Subsequently, 90 g of the sample was introduced, and the temperature of the roaster was increased to 180 °C, which required approximately 7.5 to 8 min to achieve. At this temperature, the coffee was roasted for an additional 1, 2, 3, and 4 min for the levels RL85, RL75, RL55, and RL25, respectively. Subsequently, the samples were cooled and stored in Ziploc bags.

Each coffee sample was ground on the day of analysis using a grinder (Bunn Coffee Mill, Springfield, IL, USA) set to the ‘espresso’ mode, resulting in a powder with a particle size of less than 500 µm. Prior to grinding, between 10 and 15 coffee seeds were placed in the hopper to remove any residual coffee. The extraction of the volatile compounds was performed immediately thereafter.

### 3.3. Extraction of Volatile Compounds by SPME

For the extraction of volatile compounds, a manual holder and a 2 cm, 50/30 µm PBMS/CAR/DVB fiber from Supelco (57299-U, Bellefonte, PA, USA) were employed. The fiber was conditioned at 250 °C for 60 min before each extraction. A total of 3.30 g of ground roasted coffee was weighed and placed in a 15 mL amber vial (27008, Supelco, Bellefonte, PA, USA). The vials were sealed with crimped caps and silicone headspace septa. Each vial was preheated for 10 min at 70 °C. After this period, the fiber was inserted into the vial and exposed to the headspace for 40 min at 70 °C to extract the volatile compounds [9]. All experiments were conducted in triplicate.

### 3.4. Identification of Compounds by GC-MS

For compound identification, a Thermo Fisher Scientific gas chromatograph (TRACE 1310, Waltham, MA, USA) coupled with a single quadrupole mass spectrometer (ISQ 7000, San Jose, CA, USA) was used. Due to the complexity of the sample, the identification of key aroma-active compounds was confirmed using two capillary columns: a non-polar column based on (5%-phenyl)-methylpolysiloxane (DB-5MS) and a polar column based on polyethylene glycol (DB-WAX). Both columns were 60 m in length, 0.25 mm in internal diameter, and 0.25 μm in film thickness (Agilent Technologies, Santa Clara, CA, USA). The SPME fiber containing the headspace volatile compounds was immediately inserted into the GC injection port equipped with a 78.5 mm × 6.5 mm × 0.75 mm liner (2637505, Supelco Co., Bellefonte, PA, USA) and thermally desorbed for 5 min at 200 °C using a splitless injection mode. The carrier gas, helium, was used at a constant flow rate of 2.5 mL min^−1^ (99.999% purity, Indura, Guayaquil, Ecuador). The oven conditions were programmed as described by Franca et al. [51]. The volatiles were initially identified by comparing the mass spectra of each analyte with those from standards in the NIST 2020 library. Then, the retention indices (RIs) of each compound were calculated and compared with theoretical RIs [52,53]. A C8–C22 alkane mixture (Sigma-Aldrich, Laramie, WY, USA) was used for this purpose. Data were analyzed using Chromeleon 7.3 software (Thermo Fisher Scientific, San Jose, CA, USA).

### 3.5. Identification of Aroma-Active Compounds

Prior to the analysis using the olfactometer, seven non-smoking individuals (four women and three men) were trained with the Le Nez du Café fragrance kit (NEZ001, Cassis, France), which involved recognizing and describing various odors. The selection of the panelists was based on their ability to recognize odors and the repeatability of their results. Four panelists (three women and one man) were selected. For the GC-O analysis a gas chromatograph (6890, Agilent Technologies, Palo Alto, CA, USA) was used. The chromatography conditions were the same as those employed in the GC-MS analysis described above. The column flow was split using a T-junction, dividing the flow equally between the flame ionization detector (FID) and the olfactometric detector with a sniffing port (ODP 3, Gerstel GmbH & Co. KG, Mülheim an der Ruhr, Germany). The FID was operated with a hydrogen flow rate of 30 mL/min, an air flow rate of 300 mL/min, and a detector temperature set to 220 °C. The OD detector was conditioned at 150 °C. The olfactometer was equipped with an auxiliary gas flow (humidified air) at 150 °C, designed to humidify the analyst’s nasal passages [46]. The analysts were requested to describe both the olfactory perception and the intensity of the compounds detected during the chromatographic analysis. A scale of 1 to 3 was used to describe odor intensity, with 1 being the lowest and 3 the highest. Each described odor was recorded along with the retention times and intensity. Each sample was evaluated three times by each analyst to ensure the accuracy of the results. The panelists were instructed to refrain from eating 1 h prior to analysis and to avoid wearing perfume or lotion on the day of evaluation.

### 3.6. Statistical Analysis

To determine if roasting level affects the concentration of aroma-active compounds, an analysis of variance (ANOVA) and Tukey’s multiple range test were conducted. Differences were considered significant with *p*-values ≤ 0.05. Additionally, a PCA was performed. These statistical analyses were carried out using MINITAB 16 statistical software (State College, PA, USA). Furthermore, a hierarchical cluster analysis (HCA) heat map was employed to determine the relationship between aroma-active compounds and roasting levels. R studio (version 2023.06.0+421, Posit Software, Boston, MA, USA) was used for the heat map analysis.

## 4. Conclusions

Twenty-five aroma-active compounds were identified. Of these, 22 were present across all four roasting levels, while 4-methylthiazole, furfuryl formate, and an unidentified compound (MW 46) were detected olfactometrically exclusively at the highest roasting level. Variations in aroma intensity among the aroma-active compounds were noted across the different roasting levels. At RL55 (medium roast), pleasant aromas such as sweet notes, toasted hazelnut, and caramel were observed. These aromas should be emphasized in the extraction process to enhance the final coffee aroma. In contrast, RL25 (dark roast) was characterized by undesirable aromas including burnt notes, mustiness, and sulfurous odors, which can negatively affect the aromatic quality of the coffee. Notably, the compound 2,4-dihydroxy-2,5-dimethyl-3(2*H*)-furan-3-one was described for the first time in this study and was characterized by an exceptionally strong aroma reminiscent of blackcurrant. The results will contribute to enhancing the sensory quality of coffee by optimizing the concentration of pleasant aroma compounds.

## Figures and Tables

**Figure 1 molecules-29-04723-f001:**
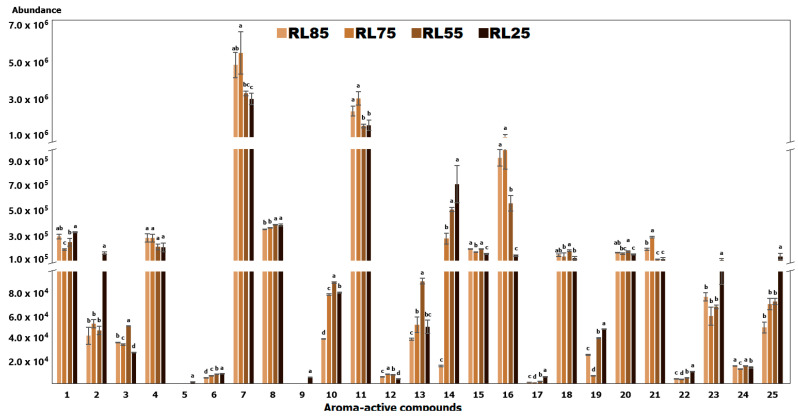
Analytical signal intensity of aroma-active compounds according to roasting level. For each compound, different lowercase letters placed above the bars indicate significant differences between the means according to ANOVA and Tukey tests (*p* < 0.05). Error bars show the standard deviation of the mean.

**Figure 2 molecules-29-04723-f002:**
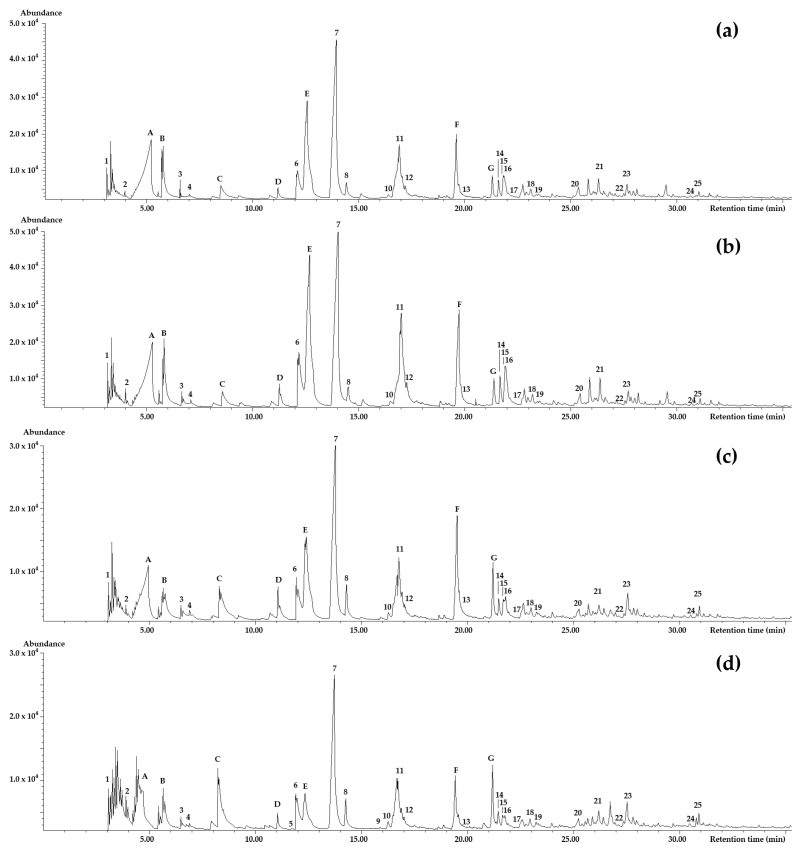
GC-FID chromatograms of coffee samples roasted at RL85 (**a**), RL75 (**b**), RL55 (**c**), and RL25 (**d**). The aroma-active compounds are labeled using numbers. Compounds that exhibit a high analytical signal but do not make a substantial contribution to the coffee aroma are designated with uppercase letters.

**Figure 3 molecules-29-04723-f003:**
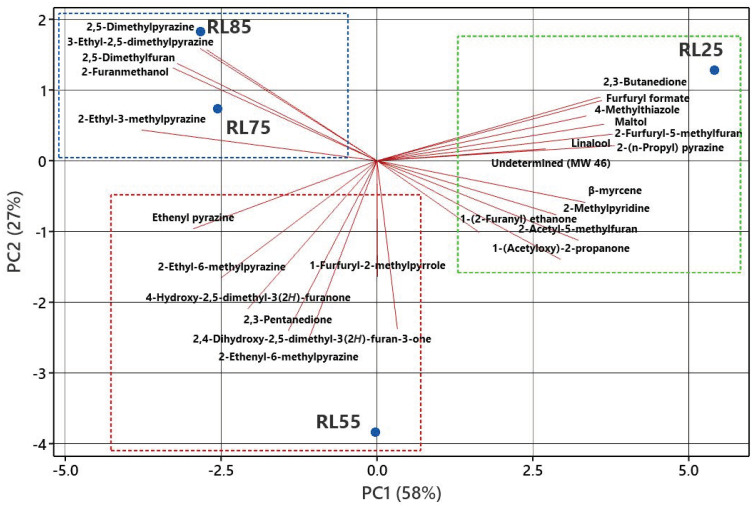
Biplot of principal component analysis of the aroma-active compounds identified in coffee seeds at different roasted levels.

**Figure 4 molecules-29-04723-f004:**
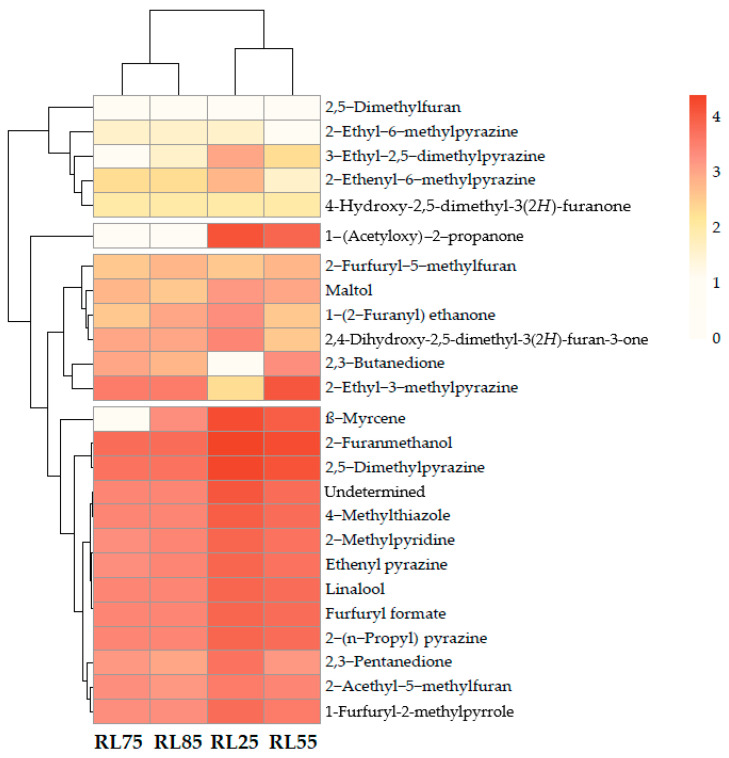
Heatmap of aroma-active compounds identified in coffee seeds at different roasting levels.

**Figure 5 molecules-29-04723-f005:**
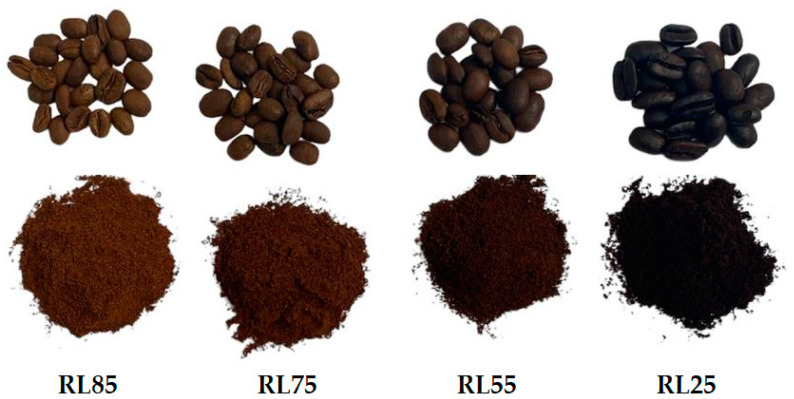
Coffee samples roast levels.

**Table 1 molecules-29-04723-t001:** Odor description and intensity of the aroma-active compounds identified in roasted coffee by GC-O-MS.

Peak No.	Compound Name	(5%-Phenyl)-methylpolysiloxane			Polyethylene Glycol			Odor Description	Odor Intensity			
		C-LRI	R-LRI		C-LRI	R-LRI			RL85	RL75	RL55	RL25
1	Undetermined (MW 46)	588	--		938	--		Rotten potato	--	--	--	3
2	2,3-Butanedione	609	599	[14]	976	977	[25]	Sweet burnt	1	1	1	2
3	2,3-Pentanedione	659	697	[14]	1057	1060	[25]	Sweet	--	--	1	--
4	2,5-Dimethylfuran	675	695	[26]	952	946	[27]	Roasted poultry	1	1	1	1
5	4-Methylthiazole	820	823	[14]	1286	1278	[27]	Stinky (unpleasant smell)	--	--	--	3
6	2-Methylpyridine	823	830	[9]	1225	1214	[27]	Sulfurous	1	1	2	3
7	2-Furanmethanol	858	868	[9]	1667	1664	[28]	Unpleasant roasted notes	2	3	2	2
8	1-(Acetyloxy)-2-propanone	870	870	[15]	1465	1477	[17]	Medicinal note	2	2	3	3
9	Furfuryl formate	906	907	[9]	1499	1497	[29]	Blackcurrant	--	--	--	1
10	1-(2-Furanyl) ethanone	908	910	[9]	1512	1512	[30]	Blackcurrant	--	--	2	2
11	2,5-Dimethylpyrazine	913	917	[9]	1328	1320	[27]	Roasted coffee	2	3	2	2
12	Ethenyl pyrazine	927	938	[9]	1445	1438	[28]	Toasted bread	1	2	2	1
13	2,4-Dihydroxy-2,5-dimethyl-3(2*H*)-furan-3-one	983	977	[31]	1562	1542	[32]	Strong blackcurrant	2	2	3	2
14	β-myrcene	993	994	[33]	1161	1156	[34]	Blackcurrant	1	2	2	3
15	2-Ethyl-6-methylpyrazine	1001	1000	[9]	1393	1390	[35]	Roasted hazelnut	2	--	2	--
16	2-Ethyl-3-methylpyrazine	1006	1004	[9]	1413	1402	[27]	Roasted nutty	2	2	1	--
17	2-(n-Propyl) pyrazine	1015	1010	[36]	1427	1428	[2]	Rubber	--	--	1	2
18	2-Ethenyl-6-methylpyrazine	1028	1038	[14]	1489	1490	[28]	Hazelnut	2	2	3	2
19	2-Acetyl-5-methylfuran	1037	1042	[9]	1627	1608	[27]	Toasted	--	--	--	2
20	4-Hydroxy-2,5-dimethyl-3(2*H*)-furanone	1071	1061	[37]	2059	2060	[37]	Caramel-like	3	3	3	3
21	3-Ethyl-2,5-dimethylpyrazine	1092	1093	[14]	1455	1449	[28]	Hazelnut toast	1	3	--	--
22	Linalool	1120	1102	[38]	1550	1552	[28]	Sweet	2	2	2	3
23	Maltol	1112	1112	[39]	1996	1981	[40]	Caramel (sweet)	2	2	2	3
24	1-Furfuryl-2-methylpyrrole	1142	1148	[26]	1832	1834	[41]	Crushed plant	1	--	1	--
25	2-Furfuryl-5-methylfuran	1159	1157	[26]	1687	1682	[10]	Green plant leaves	2	2	2	3

Calculated linear retention index (C-LRI); Reference linear retention index (R-LRI); Odor intensity: 3 (high), 2 (medium), 1 (low).

## Data Availability

Data are available from the authors upon reasonable request.

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
