# Peer review of "Effect of Roasting Level on the Development of Key Aroma-Active Compounds in Coffee"

_molecules, 2024, doi:10.3390/molecules29194723_

Round 1

Reviewer 1 Report

Comments and Suggestions for Authors

The current research investigated the effect of roasting coffee seeds on key aroma compounds in relation to roasting level. The research is well-designed and needs minor revision of some points such as 

General comments

- Coffee seeds and not beans

- It is recommended that the symbols of SI units, such as h for hours, g for gram, etc., be used and that their use be unified throughout the manuscript.

-  The full name associated with abbreviations should be mentioned with the first appearance and then the abbreviations are enough (check throughout the manuscript)

- The references are recommended to be more updated. I think the most recent one is 2021 in the references list!

Specific comments

- May you add some information of the SCA scale in brief (line 260) for the readers? indicate also the degree of roasting to the corresponding light, medium, ....., and heavily roasted and what are the relations with Melanoidins are Maillard reaction products

- Give the names of the 3 markers that appeared for the heavily roasted seeds in the conclusion (line 336).

- For 2,4-dihydroxy-2,5-dimethyl-3(2H)-furan-3-one, it has strong odor does not sense. Please, give a specific odor similar to other compounds.

- Discuss this marker (2,4-dihydroxy-2,5-dimethyl-3(2H)-furan-3-one) within the text and show its importance in quality discrimination, authenticity, ...etc. The PCA loading plot should show this marker clearly. associated with heavily roasted seeds compared with other seeds.

- The relative abundance of each compound should be also mentioned in Table 1. Also, denote that these numbers are the RI values compared with the theoretical reported values in Table 1.

- Comparisons of the relative abundances between the roasted seeds throughout the text will enrich the discussion. A comparison with previous literature is recommended.

- Classes as sub-headings are recommended for organized discussion.

- Abundances in Figure 1 can be represented as relative abundance (%).

- Where are the other chromatograms for the other 2 seeds (Figure 2)?

- Determine whether the PCA is a score plot or a loading plot

- Figure 4 title should be written as a sentence, not capitalize each word!

Comments on the Quality of English Language

No comment on the language. It is fine!

Author Response

Reviewer 1

The current research investigated the effect of roasting coffee seeds on key aroma compounds in relation to roasting level. The research is well-designed and needs minor revision of some points such as

We sincerely thank you for the valuable observations and feedback provided on our work. Your insights have significantly contributed to improving the quality and clarity of the manuscript.

General comments

- Coffee seeds and not beans

We appreciate your observation and have revised the manuscript accordingly, replacing "coffee beans" with "coffee seeds" throughout.

- It is recommended that the symbols of SI units, such as h for hours, g for gram, etc., be used and that their use be unified throughout the manuscript.

We agree with your suggestion and have ensured that SI units such as "h" for hours, "g" for grams, etc., are consistently used and correctly formatted throughout the manuscript.

-  The full name associated with abbreviations should be mentioned with the first appearance and then the abbreviations are enough (check throughout the manuscript)

The full names corresponding to abbreviations have been provided at their first appearance, followed by the abbreviations in the rest of the manuscript.

- The references are recommended to be more updated. I think the most recent one is 2021 in the references list!

We appreciate your suggestion to update the references, and we have now included more recent citations, ensuring the list is up to date with the latest relevant studies.

Specific comments

- May you add some information of the SCA scale in brief (line 260) for the readers? indicate also the degree of roasting to the corresponding light, medium, ....., and heavily roasted and what are the relations with Melanoidins are Maillard reaction products

Thank you for the suggestion. We have now briefly introduced the SCA scale and explained its relevance to light, medium, and dark roasting levels. These details can be found in lines 267-268.

- Give the names of the 3 markers that appeared for the heavily roasted seeds in the conclusion (line 336).

We have now added the specific names of the three markers associated with heavily roasted coffee seeds in the conclusion. This information is included in lines 349-351.

- For 2,4-dihydroxy-2,5-dimethyl-3(2H)-furan-3-one, it has strong odor does not sense. Please, give a specific odor similar to other compounds.

We have provided a specific odor description for this compound, indicating that it has a strong blackcurrant-like aroma. This correction has been made throughout the manuscript.

- Discuss this marker (2,4-dihydroxy-2,5-dimethyl-3(2H)-furan-3-one) within the text and show its importance in quality discrimination, authenticity, ...etc. The PCA loading plot should show this marker clearly. associated with heavily roasted seeds compared with other seeds.

We have expanded the discussion to highlight the importance of this compound in the quality discrimination and authenticity of coffee aroma, particularly in relation to its association with medium roasting levels. Nevertheless, the PCA loading plot now clearly shows the significance of this marker, which correlates with RL55 rather than heavily roasted samples.

- The relative abundance of each compound should be also mentioned in Table 1. Also, denote that these numbers are the RI values compared with the theoretical reported values in Table 1.

We appreciate your insightful comment. However, the abundances of each compound are already presented in Figure 1, and repeating this information in the table may lead to redundancy. Additionally, we chose not to use relative abundances to compare the results between roasting levels, as this could result in skewed interpretations. In relative terms, a decrease in one compound might artificially suggest an increase in another, even if its actual concentration has not increased.

- Comparisons of the relative abundances between the roasted seeds throughout the text will enrich the discussion. A comparison with previous literature is recommended.

Thank you for the suggestion. We have enriched the discussion by including comparisons of our findings with those of previous studies, particularly in the section 2.1.

- Classes as sub-headings are recommended for organized discussion.

We considered your suggestion but decided not to include sub-headings for compound classes, as our goal is to emphasize key marker compounds. However, we have reorganized the discussion to improve the flow and readability.

- Abundances in Figure 1 can be represented as relative abundance (%).

As previously noted, we prefer not to use relative abundances to compare results between roasting levels, for reasons of accuracy.

- Where are the other chromatograms for the other 2 seeds (Figure 2)?

We have now included chromatograms for the RL85 and RL75 roasting levels in Figure 2, as requested.

- Determine whether the PCA is a score plot or a loading plot

We have clarified that Figure 3 represents a biplot, as indicated in lines 217-218.

- Figure 4 title should be written as a sentence, not capitalize each word!

We have corrected the capitalization issue in the title of Figure 4, ensuring it follows proper sentence case.

Reviewer 2 Report

Comments and Suggestions for Authors

The authors made a huge work. I find the manuscript very well written. Nevetheless, scientific data could be better presented, especially statistical analysis. The Introduction section is well written and gives a proper intro to scientific importance. The experimental design and used method must be better described. The cited references are properly used and I did not find unapropriate self-citation. According all above mentioned I would like to suggest the manuscript to be accepted after minor revison.

Author Response

Reviewer 2

The authors made a huge work. I find the manuscript very well written. Nevetheless, scientific data could be better presented, especially statistical analysis. The Introduction section is well written and gives a proper intro to scientific importance. The experimental design and used method must be better described. The cited references are properly used and I did not find unapropriate self-citation. According all above mentioned I would like to suggest the manuscript to be accepted after minor revison.

Dear Reviewer,

We sincerely appreciate the time and effort you have taken to review our manuscript, and we are grateful for your valuable feedback. Your thoughtful comments have helped us to improve the clarity and quality of our work.

In response to your suggestion, we have made significant improvements in the presentation of the scientific data, particularly in the section concerning statistical analysis. We believe these changes enhance the overall understanding and rigor of the results.

Additionally, we have provided a more detailed description of the experimental design and methods, as per your recommendation. These modifications should address your concerns and improve the comprehensiveness of the manuscript.

Thank you again for your constructive review, which has undoubtedly contributed to strengthening the manuscript. We hope that the revisions meet your expectations.

Reviewer 3 Report

Comments and Suggestions for Authors

The authors provide interesting research into the key aroma compounds of coffee dependant on different roasting levels.

Line 12: the four levels should be shortly described with descriptors, e.g. light, medium, dark. The acronyms are RLxx are not universally known and only a suggestion by a trade association (no official ISO standard).

Line 19: check "burnt sweet"

Line 64: add reference to recent study

Line 89: stylistically I would add "first author et al." before [4].

Line 170: unresolved citation Flament (2001)

Line 182: are the labelled numbers the same as in the table 1? Please cross reference if appropriate

Figure 3: check, 25 marking is not visible

Figure 4: check named for correct English, e.g. hidroxy, furil

Line 242: this should read "Caturra"?

Line 297: unresovled citation Yang et al. 2021

Line 329 and throughout: please provide city/country for all manufactures/software suppliers

References: check for correct style, e.g. journal abbreviations with full stops

Author Response

The authors provide interesting research into the key aroma compounds of coffee dependant on different roasting levels.

Line 12: the four levels should be shortly described with descriptors, e.g. light, medium, dark. The acronyms are RLxx are not universally known and only a suggestion by a trade association (no official ISO standard).

We agree with your suggestion and have added brief descriptors for each roasting level, including light, medium, and dark, to clarify the acronyms RLxx. This information has been added to lines 267-268.

Line 19: check "burnt sweet"

We have corrected the term to "burnt-sweet," ensuring clarity and consistency throughout the manuscript.

Line 64: add reference to recent study

We have added an updated reference to support the information presented. You can find this in the revised manuscript.

Line 89: stylistically I would add "first author et al." before [4].

Thank you for the suggestion. We have incorporated this stylistic change.

Line 170: unresolved citation Flament (2001)

We have corrected the citation to ensure it is properly referenced in the text. Line 161.

Line 182: are the labelled numbers the same as in the table 1? Please cross reference if appropriate

We have reviewed and corrected the numbering throughout the manuscript to ensure consistency between the text and Table 1.

Figure 3: check, 25 marking is not visible

We have adjusted Figure 3 to ensure all markings, including compound 25, are clearly visible.

Figure 4: check named for correct English, e.g. hidroxy, furil

We have reviewed and corrected the spelling of compound names in Figure 4.

Line 242: this should read "Caturra"?

We have corrected the term to "Caturra" as per your observation. Line 253.

Line 297: unresovled citation Yang et al. 2021

The citation has been resolved and properly referenced in the text. Line 312.

Line 329 and throughout: please provide city/country for all manufactures/software suppliers

We have added the necessary information for manufacturers and software suppliers, as requested. You can find these details in lines 344-345.

References: check for correct style, e.g. journal abbreviations with full stops

We have reviewed and corrected the reference list, ensuring compliance with proper journal formatting standards.